# Identification and Characterization of Resistance of Three Aphid Species on Contrasting Alfalfa Cultivars

**DOI:** 10.3390/insects13060530

**Published:** 2022-06-09

**Authors:** Qiong Wu, Xiang Zhang, Xianghao Weng, Lingling Gao, Xuefei Chang, Xingxing Wang, Zhaozhi Lu

**Affiliations:** 1The First-Class Discipline of Prataculture Science of Ningxia University (No. NXYLXK2017A01), College of Agriculture, Ningxia University, Yinchuan 750021, China; 17361602775@163.com (Q.W.); kczx268@126.com (X.Z.); 2Shandong Engineering Research Center for Environment-Friendly Agricultural Pest Management, College of Plant Health and Medicine, Qingdao Agricultural University, Qingdao 266109, China; a1914208123@163.com (X.W.); changxuefei2015@163.com (X.C.); wxx0521@qau.edu.cn (X.W.); 3CSIRO Plant Industry, Private Bag 5, Wembley, WA 6913, Australia; lingling.gao@csiro.au

**Keywords:** alfalfa cultivars, antibiosis, antixenosis, EPG, cowpea aphid, pea aphid, spotted aphid

## Abstract

**Simple Summary:**

*Aphis craccivora* Koch (cowpea aphid, CPA), *Acyrthosiphon pisum* Harris (pea aphid, PA) and *Therioaphis trifolii* Buckton (spotted alfalfa aphid, SAA) are the three species of devastating pests on alfalfa in China. A study was conducted in the laboratory for identification and characterization of resistance to these three aphids among 16 of the main alfalfa cultivars planted in China. Resistance was indicated by antibiosis, antixenosis, and measuring feeding behavior using EPG (electrical penetration graph). The results indicated that different alfalfa cultivars have significantly different resistance levels to a particular species of aphid, and the same alfalfa variety also has different resistance to the three aphid species. Specifically, we evaluated the resistance of different alfalfa cultivars to CPA, which can help us for further study on the defense mechanism against CPA and for better management of this pest.

**Abstract:**

Aphids on alfalfa (*Medicago sativa*) including *Aphis craccivora* Koch (cowpea aphid, CPA), *Acyrthosiphon pisum* Harris (pea aphid, PA) and *Therioaphis trifolii* Buckton (spotted alfalfa aphid, SAA) cause significant yield losses worldwide. In this experiment, the development of these three species of aphids on 16 alfalfa cultivars was compared. The results showed that the plant cultivar had a significant influence on the development of aphids as there are significant differences in the body weight of aphids reared on different alfalfa cultivars. In addition, antibiosis between the alfalfa cultivars Pegasis and Gannong NO.9 and the three species of aphids was evaluated by measuring aphid body weight and fecundity. Antixenosis was measured using choice tests, and feeding behavior was quantified using electrical penetration graphs (EPG). The Pegasis cultivar was observed to have both antibiosis and antixenosis effects with CPA, but was susceptible to PA and SAA compared with the Gannong NO.9 cultivar. CPA had less mean body weight, less fecundity, and shorter feeding time on the Pegasis cultivar, and preferred to settle on Gannong NO.9 cultivar. In contrast, Gannong NO.9 exhibits antibiosis and antixenosis to PA and SAA compared with Pegasis, as shown by lower body weight, lower fecundity and chose to settle less often, but EPG data showed that PA and SAA showed no different significance in feeding behavior between Pegasis and Gannong NO.9.

## 1. Introduction

Insect-plant interactions are complex examples of co-evolutionary and co-adaptation processes [1]. Host plants have evolved a complex set of interdependent defence mechanisms ranging from physical barriers to the complex array of signaling molecules of the plant’s immune system leading to induced changes in the plant’s morphology, physiology and/or chemistry (producing plant secondary metabolites) to stop herbivore attack [2]. Chemical defenses can be broadly classified as antibiosis, antixenosis and tolerance [3]. Antixenosis consists in a low host acceptance, and ultimately the rejection of the plant because of physical or chemical cues that repel (or not attract) insects and alter behavioral processes involved in host acceptance [4]. Antibiosis is a type of resistance in which feeding on the plant results in alteration of insects physiological parameters (i.e., increased development rate, lower fecundity, and higher mortality rate) [4,5]. Crop cultivars incorporating one or more of these defenses host plant resistance (HPR) are an important component of integrated pest management (IPM) which aims to reduce pest numbers and crop damage through limiting settlement, feeding and reproduction of the insect pests [6,7]. HPR is generally compatible and complementary with the use of biological control and other tactics to suppress pests [8].

Insects with piercing and sucking mouthparts, such as aphids, are a serious problem for agriculture [9]. They use their stylets to penetrate the plant tissue and consume large amounts of phloem sap thus depriving the plant of photo-assimilates and limiting plant nutrients [10]. Direct damage by aphids was responsible for mean annual losses of 700,000 t of wheat, 850,000 t of potatoes and 2,000,000 t of sugar beets in Europe [11]. Alfalfa-feeding aphids are one of the most damaging insect pests on that crop, causing an estimated production loss of 25% in alfalfa (*Medicago sativa*) worldwide [12]. There are three species of aphids on alfalfa in China, *Aphis craccivora* Koch (cowpea aphid, CPA), *Acyrthosiphon pisum* Harris (pea aphid, PA) and *Therioaphis trifolii* Buckton (spotted alfalfa aphid, SAA) [13], and these cause about 20–30% yield loss of alfalfa production in China [14]. The CPA and PA are also important pests of legume herbs, such as peas, broad beans, peanuts, and soybeans [15,16]. SAA has a narrow host range and lives on alfalfa or clover (*Trifolium repens* L.) [17].

The main approach to aphid management is the application of insecticides. Using resistant cultivars might be an alternative method against aphids. Host plant resistance is a cost effective and environmentally sustainable. However, the resistance of alfalfa against the three aphid species is inconsistent. The model legume cultivar *M. truncatula* Jester is resistant to PA and SAA by the expression of independent resistance genes located in CC-NBS-LRR-rich regions on chromosome 3 [18,19,20,21]. In contrast, a CPA resistance gene was lacking in *M. truncatula* Jester, but was found in *M. truncatula* accession SA30199 on chromosome 2 [22]. Therefore, screening for aphid-resistant cultivars amongst existing germplasm resources is a simple and effective first step in identifying alfalfa cultivars that may be employed within an IPM framework for this crop.

In this study, the resistance of 16 alfalfa cultivars in China to CPA, PA and SAA is assessed using aphid development as the measure. Based on these findings, the resistance of two cultivars against the three aphid species was studied in detail by measuring aphid performance including development, reproduction, host selection and feeding behavior. Our study will inform farmers about which plant cultivars are most suited to use in the regions of China where CPA, PA and SAA exist individually or together.

## 2. Materials and Methods

### 2.1. Plants and Insects

Seeds of 16 cultivars of *M. sativa* (Table 1) were obtained from alfalfa seed companies in China: Beijing Best Grass Industry (http://www.bestseed.com.cn/, accessed on 1 June 2021), Beijing Zhengdao Seed Industry (http://mgeren3123.312green.com/, accessed on 15 May 2021), Jiuquan Daye Seed Industry (http://sp.hc23.com/company/120673.html, accessed on 15 May 2021), and Barenbrug Forage Industry (https://barenbrug.com.au/, accessed on 1 June 2021). To ensure even germination, seeds were soaked in sterile water approximately 8 h and then individually transferred to hydroponic seedling culture cotton. After germination, seedlings were placed in a growth chamber (24 ± 1 °C, 65 ± 10% RH, and a photoperiod of 14:10 h and watered with nutrient Hoagland solution. After 7 days of growth individual plants were transferred to black wide-neck-bottles (height: 8 cm, diameter: 6.5 cm) for hydroponic culture. A week later, the plants were used in experiments in climatic chambers (Hefei Youke Instrument Company Ltd., RGC-500B, in Hefei, China).

PA, SAA and CPA were collected from alfalfa crops at Guyuan, Ningxia Province and identified under microscopy according to the special characterzation of different species aphids [23], and maintained on alfalfa ‘Surprise’ (Surprise has proved to be an aphid-susceptible cultivars for three species aphids with fuzzy discernment and aphid number ratio method by Ma Jianhua [24] and Wei Shuhua (part data were not published)) in cages (35 × 35 × 70 cm) in the laboratory under the aforementioned conditions. The aphid populations were reared for several generations before experiments were conducted. The performance of the three species of aphids on different alfalfa cultivars was studied in the laboratory at 24 ± 1 °C, 65 ± 10% RH, and a photoperiod of 14:10 h.

### 2.2. Preliminary Experiment: Screening of Aphid-Resistance of Alfalfa Cultivars

To evaluate the resistance of 16 *M. sativa* cultivars to three alfalfa aphid species (PA, CPA, SAA), approximately 50 adult apterous aphids were randomly chosen from the rearing colonies and placed on a leaf surface. They were permitted to produce nymphs for 12 h and then the adult aphids and excess nymphs were removed so that each plant had six nymphs remaining. After 4 h the aphids were checked to ensure that each plant had been successfully inoculated with 6 aphid nymphs. The plants were placed on a pot tray (d = 15 cm) and covered with a cage made from a clear plastic bottle modified with a cut-off base and large mesh-covered ventilation holes to prevent aphids escaping and parasitism. After four days infestation, the PA, SAA, and CPA respectively on each plant were gently brushed off and put on the part per million weighing balance (Runlian, H0503, Xingtai, China), immediately recording the fresh aphid weight. Nine replicates were conducted for each cultivar. The weight of the same aphid grew on 16 alfalfa varieties for 4 days was used as the basis for judging alfalfa resistance aphid. It is generally considered that the higher aphid weight, the faster aphid development and the weaker of alfalfa resist to aphid. On the contrary, the lower aphid weight, the slower aphid development and the stronger resistance of alfalfa to aphid.

### 2.3. Aphid-Resistance of Two Contrasting Alfalfa Cultivars

We selected the most and least aphid-resistant alfalfa cultivars based on the results of the preliminary experiment (body weights of the different aphid species on different alfalfa cultivars during the same development period). To further characterize aphid resistance in more detail, fecundity, settling preference choice–tests and electrical penetration graph (EPG) experiments were conducted using the three aphid species on the two selected alfalfa cultivars.


*Experiment 1: Colonization ability of aphids on caged plants*


This experiment was designed to determine the ability of new nymphs to establish a colony on each of the two alfalfa cultivars. Nine individual two-week-old plants were covered with a transparent plastic cover with mesh on the top to allow plant transpiration and prevent insect escape. Two adult aphids of each of the three species were transferred to 9 individual plants respectively, and eight hours later the adults and excess nymphs were removed leaving one nymph of each species only on each plant. The number of offspring of each species on each plant was counted after 14 days.


*Experiment 2: Evaluation of antixenosis*


This experiment was conducted to identify the preference of the three aphid species for each cultivar. Four plants were arranged so that one plant occupied each of the four corners of an insect-proof cage (38 cm length × 38 cm width × 46 cm height), with two replicates of each cultivar placed diagonally. Pots were spaced so that no leaves touched other plants. A 10 cm Petri dish containing 50 aphids was placed in the center of the cage. Nine replicate cages were prepared for each aphid species. The settling of aphids on each cultivar was observed at 3, 6, 9, 24 and 48 h after release.


*Experiment 3: Electrical penetration graphs*


The feeding behaviors of the three species of aphids on the two selected resistant and susceptible cultivars were studied using the direct-current electrical penetration graph (EPG) technique with modifications. Following the manufacturer’s directions (http://www.epgsystems.eu/, accessed on 10 November 2021), a thin wire (ca. 20 μm in diameter and 5 cm in length) was carefully attached to the dorsal aspect of the thorax using a water-soluble silver glue. Each plant electrode was placed into nutrient solution in a pot. Plants were grown under 14:10 h at 24 °C. When plants were 14 days old, a single apterous adult aphid was placed on a single trifoliate leaf and the feeding behavior of the aphid was monitored. Fifteen replicates were included for each cultivar. An eight-channel amplifier (Eco Tech, EPG, Hoagland) simultaneously recorded electrical signals of eight individual aphids on separate plants, four resistant and four susceptible per day for five days. Waveform patterns were scored according to categories described by Dancewicz: non-penetration phase; pooled pathway phase activities; salivary secretion into sieve elements; phloem sap ingestion; xylem ingestion; and cell puncture events of several seconds duration (referred to as potential drop) [25]. The total duration of np (non-probing phase) indicates the level of plant antixenosis; the longer the duration, the stronger the antixenosis. E1 waveform represents salivation in phloem tissue without ingestion and E2 waveform represents secretion of watery saliva and passive ingestion in phloem sieve elements. The total duration of E1 + E2 indicates antibiosis; the longer feeding duration, the weaker the antibiosis to aphids.

### 2.4. Statistical Analysis

Data from the preliminary experiment were expressed as a mean ± standard error (SE) for the nine replicates for the three species of aphids on the 16 different alfalfa cultivars. All statistical analyses were performed using IBM SPSS Statistics v.23.0 (Chicago, IL, USA). One-way ANOVAs were used to examine the effects of alfalfa cultivar and aphid species on the response variables (weight, fecundity, settling preference and feeding time), followed by Tukey’s honestly significant difference (HSD) test. Differences between the means were tested for significance at the 0.05 confidence level. The EPG data were analyzed using the Excel Workbook for automatic parameter calculation of EPG data 4.4 [26].

## 3. Results

### 3.1. Preliminary Experiment: Weight of the Three Species of Aphids Cultured on 16 Cultivars of Alfalfa

The body weight of each species of aphid was significantly affected by alfalfa cultivar (CPA: *F* = 21.54, df = 16, *p* < 0.05; PA: *F* = 29.064, df = 16, *p* < 0.05; SAA: *F* = 13.68, df = 16, *p* < 0.05). Approximately 87.5%, 81.3% and 93.8% of tested cultivars had a moderate or high resistance to CPA, PA and SAA, respectively. Cultivars of WL series, Mufeng, Algonjin, and Sardi10 all had a relatively higher resistance to the three species of aphids, while Surprise was relatively susceptible (Figure 1A–C). (Here, the resistance level is a relative value, Lowercase letters indicate the significant differences in resistance to aphid among 16 cultivars, and the varieties not marked with letter a were all considered as moderate or high resistance level.) In addition, cultivars Gannong NO.9 and Pegasis had distinctive characteristics of resistance between CPA and PA or SAA. The CPA developed more quickly on Gannong NO.9 than Pegasis. The mean weight of CPA on Gannong NO.9 increased by 55.5% compared to that on Pegasis (*F* = 47.76, df = 1, *p* < 0.05, Figure 2A). Conversely, SAA and PA developed more slowly on Gannong NO.9 than Pegasis. The mean weight of PA and SAA on Gannong NO.9 decreased by 42.6% and 55.4%, respectively (*F* = 52.49, df = 18, *p* < 0.05, Figure 2B; *F* = 88.26, df =18, *p* < 0.05, Figure 2C).

### 3.2. Fecundity of Three Aphid Species Aphids on Caged Plants of Two Alfalfa Cultivars: Gannong NO.9 and Pegasis

The reproductive capacity of CPA cultured on the Gannong NO.9 cultivar was significantly higher than on Pegasis. The number of offsprings on Pegasis decreased by 92.4% compared with Gannong NO.9 (*F* = 224.62, df = 1, *p* < 0.05, Figure 3A). In contrast, reproduction of PA on Gannong NO.9 was significantly lower than on Pegasis 70.3% (*F* = 57.29, df = 1, *p* < 0.05, Figure 3B). There was no significant difference in the number of SAA between Gannong NO.9 and Pegasis (*F* = 2.19, df = 1, *p* < 0.05, Figure 3C).

### 3.3. The Preference of the Three Aphis Species for Settling on Plants of Two Alfalfa Cultivars: Gannong NO.9 and Pegasis

In the host selection experiment, aphids spread rapidly from the release point in the center of the cage and moved around the cage before settling. Within 24 h, the number of CPA, PA, and SAA aphids on Pegasis and Gannong NO.9 cultivars increased. The number of PA and SAA aphids on Pegasis was always greater than on Gannong NO.9. At 48 h, the abundance of PA and SAA on Pegasis and Gannong NO.9 tended to be the same (Figure 4A–C).

### 3.4. The Feeding Behavior of the Three Aphid Species on Plants of Two Alfalfa Cultivars: Gannong NO.9 and Pegasis

Based on the EPG data, there were two important waves forms np and E1 + E2. The total duration of np of PA on the Pegasis cultivar was significantly greater than on Gannong NO.9 (PA: *F* = 15.97, df = 1, *p* < 0.05), while the total duration of np of CPA and SAA was not different between Pegasis and Gannong NO.9 (CPA: *F* = 1.05, df = 1, *p* > 0.05; SAA: *F* = 0.047, df = 1, *p* > 0.05) (Figure 5A–C). The total duration of E1 + E2 of CPA on the Pegasis was significantly less than on Gannong NO.9 (E1 + E2: *F* = 22.08, df = 1; *p* >0.05), while the total duration of np of PA and SAA was not different between Pegasis and Gannong NO.9 (PA: E1 + E2: *F* = 1.87, df = 1; *p* > 0.05; SAA: *F* = 0.31165, df = 1, *p* > 0.05) (Figure 6A–C).

## 4. Discussion

There were significant differences in aphid resistance among the 16 alfalfa cultivars, and there were also differences in resistance of the same cultivar to three different species of alfalfa aphids within the first 4 days of development. Overall, according to statistics, 73.3% of the cultivars had high or moderate resistance to all three aphid species, suggesting that these cultivars could be suitable for planting in areas where the three aphids co-exist to cause serious damage to crops. About 12.5% of cultivars were relatively susceptible to the three aphid species. About 87.5%, 81.3%, and 93.8% of tested cultivars had a high or moderate resistance to CPA, PA and SAA, respectively. These resistance differences may be due to co-evolution between the hosts and aphid species [27,28]. Generally, the narrower the host range, the greater the resistance of plants to insects [29]. SAA has a relatively narrow host range compared to CPA and PA, and this species had the higher proportion of cultivars that were aphid-resistant in this study. In addition, differences in physiological enzyme activity and detoxification ability of insects on different host plants may also influence resistance to them [30,31].

Interestingly, CPA, PA and SAA had different responses on the Pegasis and Gannong NO.9. The CPA developed poorly on Pegasis compared to Gannong NO.9 as shown experimentally by less weight gain and less fecundity. In the settling choice test, Gannong NO.9 was selected by CPA in preference to Pegasis. The Pegasis cultivar exhibited greater antibiosis and antixenosis effects on CPA compared with PA and SAA. In the feeding behavior test, the duration of E1 + E2 of CPA was significantly longer on Gannong NO.9 than on Pegasis. The El and E2 wave process will be shortened if there was a resistance factor (such as chemical resistance factors) in the phloem, therefore the antibiosis effect shown by Pegasis with CPA may be explained by chemical resistance factors in the phloem, which is in accord with the index of body weight and fecundity. In contrast, PA did not prefer to settle on the Gannong NO.9 cultivar, and when it fed on Gannong NO.9 there was lower body weight and reproduction compared to when it fed on Pegasis. Plants of the Gannong NO.9 cultivar exhibited greater antibiosis and antixenosis effects on PA compared with CPA. while the np wave of PA was prolonged on the Pegasis compared with Gannong NO.9. The np wave before the 1st probing wilI be prolonged if the resistance factors were host volatilization (non-preference). Thus, Pegasis behaved antixenosis to PA, but this result was the opposite to the preference experiment. This inconsistency also has been reported previously for the performance of brown stink bug *Euschistus heros* (Hemiptera: Pentatomidae) on resistant and susceptible soybean cultivars [32], and in the pea aphid on pea cultivarsand in the soybean on soybean cultivars [33,34]. Plant resistance to aphids is a complex phenomenon. In this experiment, various cultivars of alfalfa resistant against three species of aphids were only evaluated in the laboratory, which has some limitations, and in the future complex field conditions should be considered.

According to previous investigations and statistics, there are about 85% of alfalfa cultivars among Australian and American commercial alfalfa cultivars were resistant against two or three species of aphids, including PA, SAA and BGA (blue green aphid), but there is no mention of the resistance to CPA (https://www.barenbrug.com.cn/, accessed on 10 November 2021). CPA on alfalfa was always ignored and referred to as an occasional pest. Two factors may be account for this. Firstly, historically CPA was not considered as an important pest on alfalfa and no cultivars resistance to CPA were assessed or developed in the world [35]. Secondly, there is a lack of alfalfa cultivar resources to resist CPA, and the existing known aphid resistance gene has not proved to confer resistance to CPA [36].

But now, with the warming of climate, the seasonal occurrence of CPA is no longer restricted to late autumn and early spring, as CPA was abundant in July and August [37]. In addition, the distribution of CPA has expanded north into Siberia (Russia) and Alberta (Canada) and south into Chile and Argentina (http://www.cabi.org/isc/datasheet/6192, accessed on 3 February 2022). And more importantly, there are about 20 states that have reported high population densities of CPA in alfalfa in the USA [38]. Therefore, CPA resistance should be considered in future evaluations for breeding alfalfa. Using transgenic technology or combining it with traditional breeding methods, comprehensive resistance against CPA, PA and SAA should be achieved and will benefit pest management in alfalfa production.

In our experiment, the CPA resistance was evaluated only among main planted alfalfa cultivars in China. In the future, more cultivars or germplasm from around the world needs to be screened for resistance against CPA, and also to be evaluated in the field and cropping regions. In addition, the mechanism of alfalfa resistance against CPA should be explored using omics and gene technology as a priority, as this will promote the development of pyramid cultivars with different resistant genes against various species of aphids in alfalfa.

## 5. Conclusions

Our study indicated that the defense response of alfalfa induced by CPA was different from that induced by PA and SAA. The three aphid species performed differently on different alfalfas with different resistant levels. Future studies should pay more attention to the mechanism of alfalfa resistance against CPA and promote the development of pyramid cultivars.

## Figures and Tables

**Figure 1 insects-13-00530-f001:**
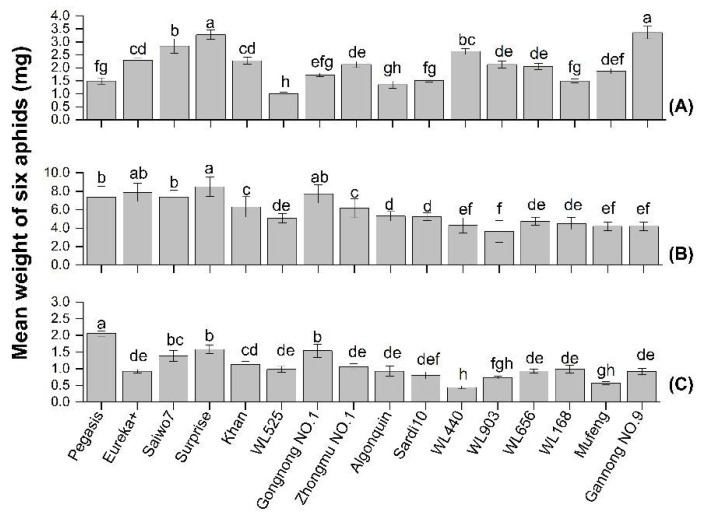
The weight of (**A**) cowpea aphid (CPA), (**B**) pea aphid (PA), (**C**) spotted aphid (SAA) on 16 alfalfa cultivars after culture for four days (Note: Different lower-case letters indicated significant differences among the cultivars (*p* < 0.05).

**Figure 2 insects-13-00530-f002:**
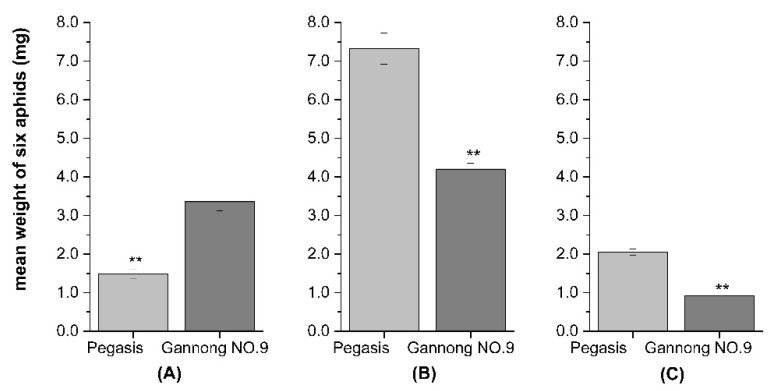
The weight of (**A**)cowpea aphid (CPA), (**B**) pea aphid (PA), (**C**) spotted aphid (SAA) on the alfalfa cultivars of Pegasis and Gannong NO.9 (data sourced from Figure 1A–C) (** Indicates that the difference is significant at *p* < 0.05).

**Figure 3 insects-13-00530-f003:**
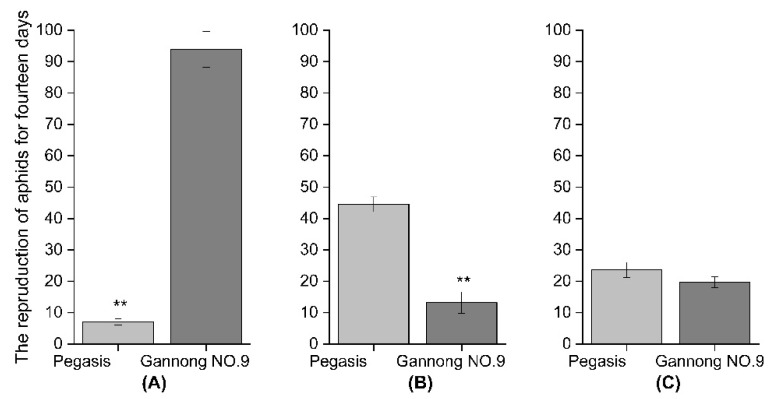
The number of offspring of (**A**) cowpea aphid (CPA), (**B**) pea aphid (PA), (**C**) spotted aphid (SAA) on the alfalfa cultivars of Pegasis and Gannong NO.9 after fourteen days (** Indicates that the difference is significant *p* < 0.05).

**Figure 4 insects-13-00530-f004:**
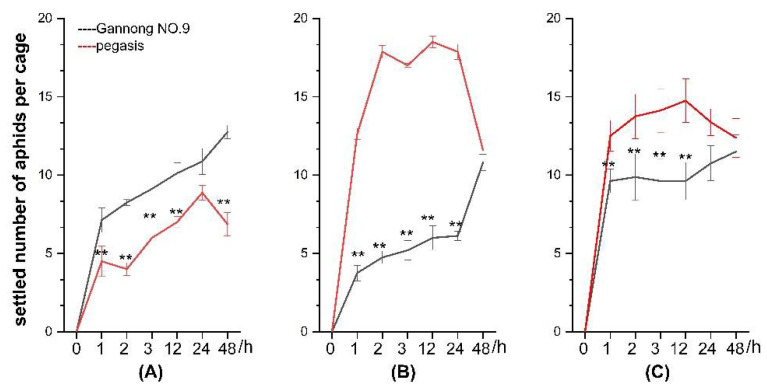
Number of (**A**) cowpea aphid (CPA), (**B**) pea aphid (PA), (**C**) spotted aphid (SAA) released and settled on the alfalfa cultivars of Pegasis and Gannong NO.9 at 1, 2, 3, 12, 24, 48 h (** Indicates that the difference is significant, *p* < 0.05).

**Figure 5 insects-13-00530-f005:**
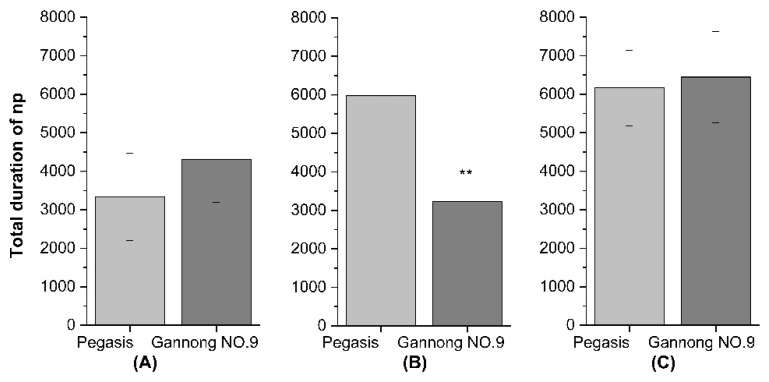
Total duration of no penetration activity (np) of (**A**) cowpea aphid (CPA), (**B**) pea aphid (PA), (**C**) spotted aphid (SAA) (** Indicates the difference is significant, *p* < 0.05).

**Figure 6 insects-13-00530-f006:**
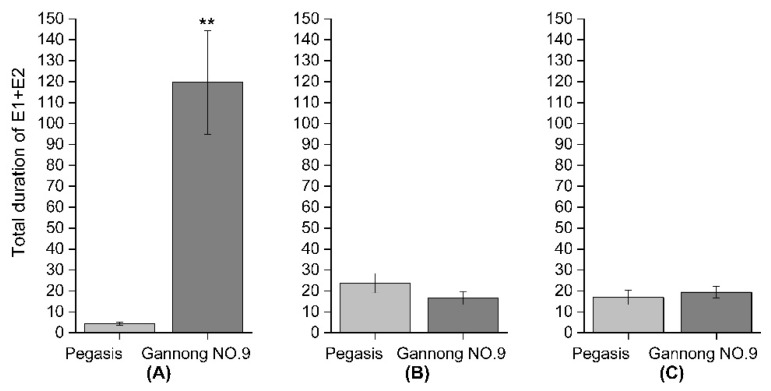
Total duration of stylet activity in phloem (E1 + E2) of (**A**) cowpea aphid (CPA), (**B**) pea aphid (PA), (**C**) spotted aphid (SAA) (** Indicates the difference is significant, *p* < 0.05).

**Table 1 insects-13-00530-t001:** The information about the tested 16 cultivars.

Number	Cultivars	Source	Number	Cultivars	Source
1	Pegasis	Australia	9	Gonggong NO.1	China
2	Eureka+	Australia	10	Zhongmu NO.1	China
3	Saiwo	Australia	11	Sardi 10	Australia
4	Surprise	Canada	12	WL440	America
5	Kehan	Canada	13	WL903	America
6	Mufeng	China	14	WL656	America
7	Algonquin	Canada	15	WL68	America
8	Gannong NO.9	China	16	WL525	America

## Data Availability

Data used in this study are available from the corresponding authors upon reasonable request. Informed consent was obtained from all subjects involved in the study.

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
