# Peer review of "Identification and Characterization of Resistance of Three Aphid Species on Contrasting Alfalfa Cultivars"

_insects, 2022, doi:10.3390/insects13060530_

Round 1

Reviewer 1 Report

General comments:

The manuscript “ Identification and characterization of resistance of three aphid species on contrasting alfalfa cultivars” provided interesting findings on host-plant resistance to key aphid pest species. The objectives were clearly described and the methods were appropriate to evaluate the relative resistance across 16 cultivars. One of the drawbacks was the lack of explanation of how resistant cultivars were rated in the different experiments. The discussion also could be more clearly written as it read in the first 2 paragraphs as a repetition of the results section with few comparisons to other literature. In some cases English language and grammar needs to be corrected.

The manuscript should be revised - several comments and suggestions are provided in the section to follow.

Specific comments:

Abstract

Lines 31-32 – Edit “ The results showed that the Pegasis cultivar behaves antibiosis and antixenosis to CPA but susceptible…” to “The Pegasis cultivar was observed to have both antibiosis and antixenosis effects with CPA, but was susceptible…”

Lines 36-37 – Edit “less body weight, less fecundity, and as the second choice to settle…” to “lower body weight, lower fecundity and chose to settle less often…”

Introduction

Lines 58-59 – Edit “Alfalfa-feeding aphids are the most damaging insect pests,” to “Alfalfa-feeding aphids are one of the most damaging insect pests on that crop,”

Materials and Methods

Line 99 – Further detail on the alfalfa “Jingxi” should be provided – is this considered an aphid susceptible variety, and is this the case for all aphid species?

Line 107 – What was the age (# days) or stage of the plants used in this experiment?

Line 135 – Did the leaves of the plants touch the bottom of the cage? How did the aphids reach the plant?

Lines 143 – 150 – Please explain in further detail how the aphid EPG was competed – were the aphids attached to the electrical signal detector? This was not clear in the description. Can you provide the website for the Eco Tech company as was done for previous seed suppliers.

Lines 160 - 164 – This description should be included in the EGP part of section 2.3 not in the statistical analysis (2.4)

Results

Lines 171-172 – It is not clear how the term “moderate resistance” or “high resistance” are being determined from the aphid weights – please explain what it is relevant to – is it the lowest aphid weight and by what factor is the difference? This should be explained in the methods section.

Figure 1 – Why were the X axis broken for A and C in order to reach 10 mg, rather than using only the highest weight as the limit? This does not seem necessary.

Figure 3 and 6 – The X axis numbering could be reduced by half (every 2nd value would be sufficient).

Discussion

Line 238 – Please provide a reference(s) for the point on co-evolution of aphids and resistant alfalfa – the following statement is not clearly explained why SAA has the greater number of alfalfa cultivars that it is resistant to based on physiological differences that were not measured in this study.

Lines 252- 252 – Edit “therefor Pegasis behaved antibiosis to CPA in phloem,”  to “therefore the antibiosis effect shown by Pegasis with CPA may be explained by chemical resistance factors in the phloem”  

Lines 254 – 255 – Edit “when fed on Gannong NO.9 this aphid had less mean body weight and less reproduction compared to feeding on Pegasis” to “when it fed on Gannong No.9 there was lower body weight and reproduction compared to when it fed on Pegasis”

Author Response

Dear reviewer,

I have revised the mannuscript in accordance with your comments,  including deleting repeated sentences, adding missing information and correcting unclear sentences.  I haved marked the important statement changes with yellow. Please check all revisions. With regards, Qiong Wu

Reviewer 2 Report

This study reports the differences in the performance and preference of three aphid species on several cultivars of alfalfa, with the results of EPG analysis. This paper is well written, and the results are clear and convincing. I think that English expression is pretty good. The authors found that suitable and unsuitable host cultivars are different among the three species.

Although I think that there are no large flaws in this paper, some minor problems remain, and in the following I will make comments on the text.

Line 136. “Conducted” should be “prepared”.

Line 171-172. Clarify what is the criteria of resistance. Which number corresponds to resistance?

Line 174. Make a space; susceptible(Figure 1A,B,C)

Line 180. Remove the period; respectively. (F = 52.49

Line 234. suggesting that ---

Line 239-240. Cite reference.

Line 250-251. This sentence may be wrong. “prolonged” may be “shortened”.

Line 261. Italicize “Euschistus heros”

Line 262. “in pea to pea aphid” may be “in the pea aphid on pea cultivars”. “In soybean to soybean aphid” may be “in the soybean aphid on soybean cultivars”.

Line 274. “to resistant CPA” may be “to resist CPA”.

Line 280. “important” may be “importantly”.

Author Response

(The authors gave the same response as above.)

Reviewer 3 Report

The manuscript number titled “Identification and Characterization of Resistance of Three Aphid Species on Contrasting Alfalfa Cultivars” authored by Wu et al., describes host plant resistance in alfalfa against three most devastating aphid species. They found that some commercial alfalfa cultivars have moderate resistance against aphids with potential application of these cultivars by farmers. Minor comments and suggestions are in blue in the attached pdf file.

Author Response

(The authors gave the same response as above.)

Round 2

Reviewer 1 Report

The revised manuscript has been reviewed and most of the original concerns have been addressed and the manuscript improved. One remaining concern is as follows:

Line 99 (version 1) – Further detail on the alfalfa “Jingxi” should be provided – is this considered an aphid susceptible variety, and is this the case for all aphid species?

The authors have edited the information about the susceptible alfalfa variety from “ PA, SAA and CPA were … maintained on alfalfa ‘Jingxi’ in cages (35×35×70 cm) in the laboratory” to “PA, SAA and CPA were … identified under microscopy according to the special characterzation of different species aphids[23],and maintained on alfalfa ‘Surprise’ an aphid-susceptible cultivars for three species aphids, pre-experiments have proved itsin cages (35×35×70 cm) in the laboratory….” (Lines 105-106 version 2)

The latter statement regarding susceptible variety did not include a reference or information on how the susceptibility was rated prior to this experiment (the sentence seems to be incomplete) – The authors should provide a edited sentence and reference if possible to correct this.

Author Response

Dear reviewer:

Thanks very much for taking your time to review this manuscript. We really appreciate all your professional comments and suggestions. We have studied comments carefully and have made correction which we hope meet with approval. Revised portion has been marked in green in the manuscript.

With regards.

 Qiong Wu
